# Performance of Expanded Newborn Screening in Norway Supported by Post-Analytical Bioinformatics Tools and Rapid Second-Tier DNA Analyses

**DOI:** 10.3390/ijns6030051

**Published:** 2020-06-27

**Authors:** Trine Tangeraas, Ingjerd Sæves, Claus Klingenberg, Jens Jørgensen, Erle Kristensen, Gunnþórunn Gunnarsdottir, Eirik Vangsøy Hansen, Janne Strand, Emma Lundman, Sacha Ferdinandusse, Cathrin Lytomt Salvador, Berit Woldseth, Yngve T. Bliksrud, Carlos Sagredo, Øyvind E. Olsen, Mona C. Berge, Anette Kjoshagen Trømborg, Anders Ziegler, Jin Hui Zhang, Linda Karlsen Sørgjerd, Mari Ytre-Arne, Silje Hogner, Siv M. Løvoll, Mette R. Kløvstad Olavsen, Dionne Navarrete, Hege J. Gaup, Rina Lilje, Rolf H. Zetterström, Asbjørg Stray-Pedersen, Terje Rootwelt, Piero Rinaldo, Alexander D. Rowe, Rolf D. Pettersen

**Affiliations:** 1Norwegian National Unit for Newborn Screening, Division of Paediatric and Adolescent Medicine, Oslo University Hospital, 0424 Oslo, Norway; insaev@ous-hf.no (I.S.); jejorgen@ous-hf.no (J.J.); erkris@ous-hf.no (E.K.); jstran@ous-hf.no (J.S.); emlund@ous-hf.no (E.L.); carsag@ous-hf.no (C.S.); oeols@ous-hf.no (Ø.E.O.); mocber@ous-hf.no (M.C.B.); ankjaa@ous-hf.no (A.K.T.); andzie@ous-hf.no (A.Z.); jinzha@ous-hf.no (J.H.Z.); likaso@ous-hf.no (L.K.S.); maytre@ous-hf.no (M.Y.-A.); silhog@ous-hf.no (S.H.); sivloe@ous-hf.no (S.M.L.); molafsen@ous-hf.no (M.R.K.O.); dionav@online.no (D.N.); junitagaup@hotmail.com (H.J.G.); astraype@ous-hf.no (A.S.-P.); alerow@ous-hf.no (A.D.R.); rdpetter@ous-hf.no (R.D.P.); 2Department of Paediatrics, University Hospital of North Norway, 9019 Tromsø, Norway; Claus.Klingenberg@unn.no; 3Paediatric Research Group, Department of Clinical Medicine, UiT The Artic University of Norway, 9019 Tromsø, Norway; 4Department of Paediatrics, Division of Paediatric and Adolescent Medicine, Oslo University Hospital, 0424 Oslo, Norway; gunngu@ous-hf.no (G.G.); rlilje@ous-hf.no (R.L.); trootwel@ous-hf.no (T.R.); 5Department of Paediatrics, Haukeland University Hospital, 5021 Bergen, Norway; eirik.vangsoey.hansen@helse-bergen.no; 6Laboratory Genetic Metabolic Diseases, Department of Clinical Chemistry, Amsterdam University Medical Centers, University of Amsterdam, AZ 1105 Amsterdam, The Netherlands; s.ferdinandusse@amsterdamumc.nl; 7Norwegian National Unit for Diagnostics of Congenital Metabolic Disorders, Department of Medical Biochemistry, Oslo University Hospital, 0424 Oslo, Norway; catsal@ous-hf.no (C.L.S.); bwoldset@ous-hf.no (B.W.); ybliksru@ous-hf.no (Y.T.B.); 8Centre for Inherited Metabolic Diseases, Karolinska University Hospital, Solna, Sweden, Department of Molecular Medicine and Surgery, Karolinska Institutet, SE-171 76 Stockholm, Sweden; rolf.zetterstrom@sll.se; 9Institute of Clinical Medicine, University of Oslo, 0318 Oslo, Norway; 10Biochemical Genetics Laboratory, Department of Laboratory Medicine and Pathology, Mayo Clinic, Rochester, NY 55902, USA; Rinaldo@mayo.edu

**Keywords:** newborn screening, dried blood spots, cut-off values, CLIR, second-tier DNA testing, outcome

## Abstract

In 2012, the Norwegian newborn screening program (NBS) was expanded (eNBS) from screening for two diseases to that for 23 diseases (20 inborn errors of metabolism, IEMs) and again in 2018, to include a total of 25 conditions (21 IEMs). Between 1 March 2012 and 29 February 2020, 461,369 newborns were screened for 20 IEMs in addition to phenylketonuria (PKU). Excluding PKU, there were 75 true-positive (TP) (1:6151) and 107 (1:4311) false-positive IEM cases. Twenty-one percent of the TP cases were symptomatic at the time of the NBS results, but in two-thirds, the screening result directed the exact diagnosis. Eighty-two percent of the TP cases had good health outcomes, evaluated in 2020. The yearly positive predictive value was increased from 26% to 54% by the use of the Region 4 Stork post-analytical interpretive tool (R4S)/Collaborative Laboratory Integrated Reports 2.0 (CLIR), second-tier biochemical testing and genetic confirmation using DNA extracted from the original dried blood spots. The incidence of IEMs increased by 46% after eNBS was introduced, predominantly due to the finding of attenuated phenotypes. The next step is defining which newborns would truly benefit from screening at the milder end of the disease spectrum. This will require coordinated international collaboration, including proper case definitions and outcome studies.

## 1. Introduction

An application for an expanded newborn screening (eNBS) program in Norway was presented to the Norwegian Directory of Health in 2006. Following a Health Technology Assessment in 2007, the Screening Committee’s recommendations for eNBS were published in 2009. In March 2012, the Norwegian NBS program was expanded to include 23 disorders following a parliamentary decision and a revision of the regulations on the mass genetic screening of newborns. In addition to congenital hypothyroidism (CH) and phenylketonuria (PKU), screening for which was introduced in the 1970s, NBS for cystic fibrosis (CF), congenital adrenal hyperplasia, and 19 other inborn errors of metabolism (IEMs) were implemented in March 2012 (www.lovdata.no). Screening for 3-hydroxy 3-methylglutaryl-CoA (HMG-CoA) lyase deficiency was added in 2018 as part of a further expansion of the program that also included screening for severe combined immunodeficiency (SCID). The initial expansion of the NBS program in 2012 was started without any prior pilot projects to guide the algorithms for cut-offs based on screening percentiles. Thanks to a near-identical panel of NBS disorders and a similar choice of instrumentation and liquid chromatography tandem mass spectrometry (LC-MS/MS) kits as in the Swedish NBS program, we were able to launch our program by adopting their cut-off values. The post-analytical software tool Region 4 Stork (R4S) [1,2] was included in the NBS algorithm after personnel training at the Mayo Clinic, Rochester, USA. We also developed the direct second-tier DNA sequencing of the original screening samples as a further confirmatory method. The timeline for NBS in Norway is shown in Figure 1. Both NBS and the subsequent confirmatory biochemical analyses for IEMs in Norway are organized as a national service at the Oslo University Hospital (OUH, National Unit for Newborn Screening and Advanced Laboratory Service for Metabolic diseases). The NBS program in Norway is governed by general legislation covering all national specialist health care services in Norway and a specific regulation on the mass genetic testing of newborns. An advisory board—consisting of one patient representative, two geneticists, one health care provider from each of the four health regions in Norway and the director of the Norwegian Biotechnology Advisory Board—ensures that the services provided by the NBS unit fulfill the national mandate. The annual report is reviewed by the Norwegian Directorate of Health. The annual reports and evaluations of the advisory board are made publicly available (in Norwegian only) at https://forskningsprosjekter.ihelse.net/senter/rapport/L-OUS-16/2019. Participation in the NBS program is voluntary and based on informed, but not written, consent. Parents have the option to decline participation in the NBS program and can also refuse the storage of the screening sample. The latter requires a written confirmation. Using NBS samples for research purposes generally requires permission from the Regional Committees for Medical and Health Research Ethics and additional parental written consent. 

The objective of this paper is to describe the screening results, experience with second-tier mass spectrometry methods and DNA testing, and the clinical outcomes and challenges experienced during the first eight years after expanding our NBS program. To assess the development of our data interpretation methods, we re-analyzed all the samples, replicating the tools created with R4S as closely as possible using its successor—Collaborative Laboratory Integrated Reports 2.0 (CLIR, https://clir.mayo.edu). We explored the ability of these in silico tools to accurately discriminate between true and false-positive cases directly from the primary NBS data and present the results of these for all of our cases.

## 2. Materials and Methods

### 2.1. Study Population and Definitions

As the overall participation in NBS in Norway has been very close to 100%, the number of newborns screened was extrapolated from the total number of live births in Norway in the same period (2012–2020). The number of children born in Norway in the decade before eNBS (2002–2012), as well as that during the eNBS period (2012–2020), was obtained from the Medical Birth Registry of Norway [3]. Three pediatricians and a clinical geneticist employed at the NBS unit evaluate the results and report positive screening results to patients and/or pediatricians at the infant’s local hospital. The same physicians also participate in the clinical follow-up of the majority of the IEM patients identified. Disease-specific protocols for confirmatory diagnosis, treatment and follow-up have been developed for all screening conditions, to guide the treating physicians, and are available online at https://oslo-universitetssykehus.no/avdelinger/barne-og-ungdomsklinikken/nyfodtscreeningen/nyfodtscreening#behandlingsprotokoller. Written feedback on the clinical outcome and the date of closure in the case of an false positive (FP)Fresult is routinely requested from the consultant pediatrician in charge at the local hospital for all reported screening positive cases. While the PKU treatment program in Norway is organized as a national service with the lifetime follow-up of all patients at OUH, the clinical care for patients with other IEMs is not formally centralized. However, in practice, more than 90% of patients with a confirmed metabolic diagnosis from eNBS are followed up by pediatricians employed at the NBS unit, either exclusively or in collaboration with their local pediatric departments. In the remaining minority of patients (<10%), close collaboration with the treating pediatricians and written feedback reports have enabled the monitoring of patient outcomes.

Screening results and confirmatory metabolic diagnostics were readily available from the hospital laboratory information system (UNILAB, Alphasoft GmbH) and from patient registries for NBS and diagnostic follow-up. Supplementary clinical information for screening positive patients was obtained from medical records and feedback reports from local hospitals. FP cases were either defined as positive screening cases referred to a pediatric department who received normal results on the follow-up metabolic tests or cases that were designated as “biochemical” diagnoses without clinical consequences (i.e., normal enzyme activity). Maternal Inborn Errors of Metabolism (IEMs) detected through children’s screening tests were not included in the FP/TP score. The time from birth to diagnosis was defined as the days from birth to biochemical confirmation or, in cases where biochemical follow-up tests were inconclusive or preceded by genetic results, the date of the reporting of either genetic or enzymatic confirmation. The time from birth until the resolution of an FP screening result was defined as the days from birth to the date when the final confirmatory test results were communicated to the parents. In cases where further follow-up appointments were needed (e.g., maternal vitamin B12 deficiency), the final outpatient consultation was chosen as the date for the closure of the FP case. Neuropsychological/cognitive testing was performed only when requested by the physician(s) in charge. Severe clinical outcomes were defined as infants and children with global developmental delay, organ failure or death. Mild to moderate outcomes were defined as cases with developmental delay such as the delayed acquisition of milestones or mildly to moderately abnormal findings from neuropsychological tests. Good health outcomes were defined as cases where no developmental delay was observed during clinical follow-up. This project was approved by the IRB at the Oslo University Hospital June 7th 2017 (2017/2879). Written informed consent was obtained from the parents of children with metabolic conditions where less than 5 cases were ascertained and/or in cases given a clinical description in the paper. 

### 2.2. Newborn Screening Methods

Capillary (or venous) blood samples were collected on filter cards 48–72 h after birth and sent by prioritized mail to the Norwegian National NBS laboratory. No national or regional IT infrastructure is yet available to record the existence or status of NBS samples prior to arrival at the NBS department. Our ability to identify missing or lost samples depends entirely on local maternity wards maintaining complete records of screened babies and cross-checking the written screening results sent by the NBS department.

#### 2.2.1. First-Tier Methods

Amino acids, acylcarnitines and succinylacetone were extracted from a single 3.2 mm-diameter punch from each dried blood spot (DBS) using the NeoBase Non-derivatized kit and, from 2019, the NeoBase 2 Non-derivatized kit (PerkinElmer, Turku, Finland) and quantified by flow injection analysis with ultra-performance liquid chromatography coupled to tandem mass spectrometry (UPLC-MS/MS). These first-tier analyses were performed on two Acquity Xevo-TQS systems, an Acquity Xevo TQS micro or a Quattro Premier XE (Waters, Milford, MA, USA). The first-tier cut-off values used for the 20 IEMs are depicted in Table 1. In the case of an abnormal screening result in the first assessment, two new 3.2 mm diameter DBS punches were re-analyzed.

Biotinidase activity in DBSs was initially analyzed with a Victor Multilabel Plate Reader (PerkinElmer, Turku, Finland) and measured by a semi-quantitative method using biotin-6-amidoquinoline as a substrate [4]. From 2013, screening for biotinidase deficiency (BD) was performed using the Genetic Screening Processor (GSP^®^) and the GSP Neonatal Biotinidase kit, both from PerkinElmer.

#### 2.2.2. Second-Tier Methods

In cases where the duplicate repeat analyses still returned an abnormal screening result, second-tier methods were used to clarify these results. Pivaloylcarnitine is a pivmecillinam metabolite isobaric to isovalerylcarnitine (C5) and the cause of many false-positive results in first-tier screening for isovaleric acidemia (IVA). Samples with a C5 concentration >1 µmol/L were therefore tested for the chromatographic separation of pivaloylcarnitine from C5 using an in-house developed LC-MS/MS method. A second-tier test to quantify allo-isoleucine was introduced in December 2017 to improve the screening for maple syrup urine disease (MSUD), based on the method of Alodaib et al. [5]. The first-tier screening method does not differentiate between allo-isoleucine, leucine, isoleucine and hydroxyproline. Hence, in samples with either isoleucine/leucine (Xle) >250 µmol/L or valine >250 µmol/L and the ratio of Xle/alanine > 1.5, a second-tier analysis for MSUD was performed. The LC-MS/MS method implemented in the laboratory separates and quantifies the four branched amino acids allo-isoleucine (pathognomonic marker for MSUD), isoleucine, leucine and valine. Hydroxyproline is separated but not quantified.

Since December 2018, two LC-MS/MS methods to measure total homocysteine (tHcy), methylmalonic acid and methylcitric acid have been implemented as second-tier tests from DBSs. These methods were adapted from Fu et al. [6]. For other disorders, second-tier DNA sequencing was used to seek to resolve abnormal first-tier results.

#### 2.2.3. Second-Tier DNA Sequencing

Conventional Sanger sequencing on DNA extracted from the original DBSs was gradually introduced, beginning in 2012 (Table 1). This strategy was first used for medium chain acyl-CoA dehydrogenase deficiency (MCADD), very long-chain acyl-CoA dehydrogenase deficiency (VLCADD) and holocarboxylase synthetase deficiency (HCS). The DNA extraction method from the NBS DBSs has previously been described [7,8]. C5OH levels above the cut-off (>0.85 µmol/L) with second-tier genetic testing for HCS was implemented from October 2012. Thereafter, C5OH values above the cut-off were not reported if the Sanger sequencing of *HLCS* was negative (wild-type or carrier status), in order to avoid reporting 3-methylcrotonyl-CoA carboxylase (3-MCC) deficiency, a condition not included among our eNBS disorders. Meanwhile, the sequencing of *HMGCL*, the gene associated with HMG-CoA lyase deficiency, was added to the algorithm for samples with C5OH values above the cut-off from January 2018. Sanger sequencing was performed using an Applied Biosystems 3500xL Dx Genetic Analyzer, with analysis performed using the Variant Reporter software (Thermo Fisher Scientific Inc, Waltham, Massachusetts, USA). The primer sequences are available upon request. Sequencing to determine the phase of recessive variants was performed where necessary upon the receipt of parental DNA samples, also delivered as DBSs. Next generation sequencing (NGS) with amplicon-based gene panels using DNA extracted from the original DBSs was introduced for the rapid confirmatory sequencing of multi-gene conditions such as methylmalonic aciduria (MMA), propionic aciduria (PA), MSUD, long-chain-3-hydroxy acyl-CoA dehydrogenase deficiency (LCHADD)/trifunctional protein deficiency (TFP) and multiple acyl-CoA dehydrogenase deficiency (MADD), starting in 2016. The Ion AmpliSeq library kit was used with the Thermo Fisher predesigned gene panel IEMv1, which was sequenced on a benchtop ION-PGM (Thermo Fisher Scientific, CA, USA). The annotated variant calling file (vcf) was filtered in the Ion Reporter^™^ Software to show only variants in the genes relevant for eNBS disorders. The BAM files were visualized in Integrative Genomics Viewer (IGV) [9] and Alamut Visual (v.2.11, Interactive Bioinformatics, France). Variant evaluation was performed according to the American College of Medical Genetics (ACMG) guidelines [10]. The assumed pathogenic gene variants identified by NGS were always confirmed using Sanger sequencing, and after reporting the positive screening results, the segregation testing of the parents was performed either by the NBS unit or by referral to another clinical genetic laboratory. In a further development, we used the Ion AmpliSeq On-Demand pipeline (Thermo Fisher Scientific) for the customized design and synthesis of an eNBS-dedicated multiplexed gene panel. As of 2020, the amplicon-based NGS panel included all IEMs in the eNBS and their differentials plus genes for the United States of America’s recommended uniform screening program (RUSP) for disorders such as SCID and CF. Primers were designed to provide amplicons (average 200 bp) with 99% coverage of the coding sequence and a minimum of 10 bp of the flanking regions of associated introns. This eNBS-NGS panel was specially designed to be used in our nationwide screening as second-tier DNA testing based on the original DBSs. The eNBS-NGS gene list is available upon request. All variants were interpreted in the light of the biochemical data. We only reported variants likely to be causative of the aberrant first-tier biochemical screening results. Variants of uncertain significance (VUS) in genes connected to disorders fitting with the biochemical findings were reported if they fulfilled some, but not all, evaluation criteria to be classified as pathogenic (VUS+/ACMG3+) (www.acgs.uk.com). Carriers were not reported.

### 2.3. Diagnostics

As a result of a positive screening call, supplementary diagnostic biochemical tests were requested according to published protocols for each disorder. The in-house national diagnostic laboratory performed confirmatory assays including analyses of plasma amino acids and acylcarnitines, urinary organic acids and carnitine in both plasma and urine. Biotinidase activity was measured in the serum [11], and carnitine-palmitoyl-transferase-II (CPT- II) enzyme activity, in leukocytes [12]. The latter method was used to distinguish CPT-II deficiency from carnitine acylcarnitine translocase (CACT) deficiency until the rapid sequencing of DNA from DBSs became available. Other enzyme analyses were done at the Laboratory of Genetic Metabolic Diseases, Amsterdam UMC, the Netherlands (www.labgmd.nl). Residual enzyme activities (in lymphocytes) were expressed as percentages of the means of healthy controls [13]. Leucine [1-14C] decarboxylation rate assays were done in cultured fibroblasts at Centro de Diagnostico de Enfermedades Moleculares in Madrid, Spain (www.cbm.uam.es/cedem). Experimental plasma cystathionine-β-synthase activity was assayed upon personal request by Prof. Viktor Kožich, Charles University in Prague [14,15]. For MMA and PA, complementation analysis/in vitro B12 responsiveness measurement and the measurement of propionate incorporation in fibroblasts, respectively, were undertaken at the Metabolic Laboratory, Division of Metabolism, University Children’s Hospital Zürich.

## 3. Results

Between March 1st 2012 and February 29th 2020, 461,369 children were screened for 20 IEMs in addition to PKU. The PKU data are not reported here. Screening for HMG-CoA lyase deficiency was added on January 1st 2018, and by 29th of February 2020, 123,500 had been screened for this disorder. In the period 2012–2020, 182 abnormal results for the included IEMs were reported (1:2534). Positive NBS samples were collected at a median age of 53 h (range: 40–87 h), and the findings, reported at a median age of 6 days after birth (range: 2–27 days). Seven cases with conditions not prone to early presentation (partial biotinidase deficiency (BD) (*n* = 3), carnitine transporter deficiency (CTD) (*n* = 3), and cystathionine β-Synthase (CBS) deficiency (*n* = 1)) were reported more than 14 days after birth (due to verification testing and the turnaround time of DNA analyses). Among them, three were true-positive cases (one partial BD, one CTD and one pyridoxine responsive CBS deficiency). Seventy-five true-positive (TP) cases were identified (1:6151): 39 fatty acid oxidation (FAO) defects, 27 organic acidurias (OA) and 9 aminoacidopathies (AA), (Table 2). One patient was diagnosed with both trifunctional protein deficiency (TFP) and mild PKU. The post-NBS diagnostic confirmation of TP cases took a median of 8 days (2–704 days). In a few cases, the diagnostic process was time- consuming due to the processing of fibroblast cultures followed by enzyme analysis (CPT-I and MSUD). In an extreme case (transient riboflavin responsive MADD), the final diagnosis was eventually confirmed after 704 days by a research project [16]. During 2012–2014, the median time to diagnostic confirmation was 9 days (2–704 days), compared to the median of 7 days (3–31 days) in the years 2015–2020. Sixteen (21%) of the 75 newborns with TP screening results were symptomatic at the time of the NBS result (Table 3). The NBS result was the first diagnostic indication in eleven symptomatic newborns, whereas five newborns (IVA (n = 1)—PA (n = 2), MMA (n = 1) and TFP (n = 1))—were diagnosed by targeted diagnostic testing before the NBS result was available. Overall, 107 (59%) of the reported cases were FPs (1:4311). The vast majority of FP cases (73%) were positive for either low free carnitine or elevated propionylcarnitine (Table 2). Forty-nine (45%) of the total FP cases were reported during the first 10 months of the eNBS program in 2012. The median age for clarifying false-positive screening results (available for 99/107 (92%) of the FP cases) was 27 days (4–369). A FP case with benign hyper-methioninemia was followed-up for 369 days. During the two first years of the screening program (2012–2014), FP cases were resolved at a median 27.5 days of age (4–374) compared to the median of 19 days (5–222) between 2015 and 2020.

Three asymptomatic mothers (CTD (n = 2) and 3-MCC deficiency (n = 1)) were revealed through their screening-positive children. Five false-negative (FN) screening cases were later identified by targeted diagnostic testing: intermittent MSUD (n = 2), CTD (n = 2) and CPT-II (n = 1). The overall positive predictive value (PPV) excluding PKU was 40% over the entire 8 year period, with the PPV increasing from 26% in 2012 to 54% in 2019. The overall FP rate was 0.025%. Thirty-two TP cases (43%) had immigrant parents. Confirmatory testing post-NBS beyond genetic analysis, such as overall mitochondrial β-oxidation analyses, biotinidase enzyme activity measurement, complementation analysis and enzyme analysis, was performed in 45/75 (60%) of TP cases (Table 4) and in 19/107 (18%) of FP cases. Fibroblasts were obtained from seven TP infants to confirm a diagnosis or to characterize their phenotype severity (MADD/GA-II (n = 1), MMA (n = 2), PA (n = 1), MSUD (n = 1), CPT-I (n = 1) and CACT (n = 1)) or to exclude disease in four FP cases (MSUD (n = 2), 3-MCC (n = 1) and MADD/GA-II (n = 1)). Overall, 604 samples underwent rapid DNA analysis from the original DBSs. The majority were Sanger sequenced due to ambiguous/borderline cut-off biochemical screening results, and the sequence results were ready for interpretation within 1–3 working days. Seventy-four (98.7%) of TP cases were molecular verified, and of these, 57/75 (76%) were first confirmed by the rapid DNA analysis of the original filter-card samples. The proportion of molecular confirmations on DNA extracted from the original DBSs increased from 10/23 (43%) in 2012–2014 to 48/51 (94%) in the period between 2015 and 2020. Based on the DNA analyses, 513 samples were declared to be normal despite a slightly abnormal biochemical screening result, since no pathogenic variants or only one pathogenic variant was identified in the relevant genes for disorders following an autosomal recessive inheritance. After reporting a positive screening result based on DNA analysis, parental segregation testing further allowed for the identification of *cis* and *trans* alleles and exclusion of allele drop-out and assumed deleterious variants located in *cis*.

### 3.1. Targeted Diagnostic Screening in the Decade before NBS

Of the 596,591 children (<18 years of age) born in the decade before newborn screening (2002–2012), 52 patients (1:11,470) were diagnosed clinically during 2002–2020 with IEMs included in the eNBS panel (Table 2). The median (range) age at diagnosis was 243 days (2–3109 days). A single infant with HMG-CoA-lyase deficiency was diagnosed at seven months of age (in 2015, three years before the condition was part of eNBS). The incidence of patients with IEMs included in the eNBS panel increased by 46% after eNBS was introduced compared to clinically presenting patients in the preceding decade.

### 3.2. Clinical Outcomes

Four patients (5.3%) identified by eNBS were lost to follow-up after their families moved out of Norway (MCADD (n = 3), and CTD (n = 1)). With these excluded, 58/71 (82%) had good health outcomes. For the remaining 13 patients, three neonatal TFP patients (two siblings) died before 6 months of age due to severe cardiomyopathy. Three patients with PA (including a patient that had received a liver transplant due to severe recurrent hyperammonemic crises) and a single patient with neonatal CPT-II deficiency had global developmental delay. Two patients with GA-I developed acute and insidious onset neurological sequelae, respectively. A patient with MMA received a kidney transplant at 4 years of age due to the development of end-stage renal failure. Three other TP cases were categorized with mild to moderate development delay (MSUD (n = 1), MMA (n = 1) and MADD/GA-II (n = 1)). The five false-negative (FN) cases showed normal development.

#### 3.2.1. Fatty Acid Oxidation Defects

Of the 27 cases reported with possible CTD, two asymptomatic maternal cases and three newborns were diagnosed with CTD biochemically and by molecular testing (Table 1, Table 2 and Table 4). In the single case with CPT-IA deficiency, plasma acylcarnitine analysis failed to confirm the diagnosis (free carnitine, 12 µmol/L). A diagnosis of CPT-IA deficiency was established by enzyme analysis in fibroblasts and by Sanger sequencing after 74 days (Amsterdam UMC, the Netherlands) (Table 4). In the three male newborns with myopathic CPT-II deficiency, (C16 + C18:1)/C2 was the only informative screening marker (Table 1). A summary of genetic testing (with at least one allele associated with a myopathic CPT-II phenotype) is presented in Table 4. During episodes of common infections, myopathic CPT-II cases showed a transient elevation of creatinine kinase (CK) levels (range: 1500–8000 U/L) that returned to normal levels after the episodes. The only child with the severe neonatal form of CPT-II presented with multiple congenital malformations at birth and developmental arrest.

Twenty-one MCADD screening-positive cases were reported; 18 were biochemically and molecularly confirmed (PPV 86%). One case was homozygous for the mild allelic variant NM_000016.4(ACADM):c.199T>C and had residual enzyme activity of 86%. The infant was categorized as healthy and was released from further follow-up. Ten (59%) MCADD cases were homozygous for NM_000016.4(ACADM):c.985A>G (Table 4). A single MCADD patient had symptoms before the NBS result (Table 3). The MCADD incidence increased from 1:74,573 in the decade before eNBS to 1:27,139 after the implementation of eNBS. Patients diagnosed after clinical presentation and born in the decade 2002–2012 were all homozygous for the common mutation (c.985A>G). None of the MCADD patients detected in this 18-year period experienced new metabolic crises after diagnosis.

VLCADD was confirmed in six of 11 screening-positive reported cases. The R4S/CLIR tool “VLCADD dual scatter plot” was applied for the six TP VLCADD cases, and four were classified as “VLCADD” and the other two, as “not informative to differentiate between VLCADD and VLCADD carrier”. Five infants diagnosed with VLCADD showed 7–17% residual enzyme activity (rEA). Four of the five children were either compound heterozygous or homozygous for the common mild allelic variant c.848T>C (Table 4), and all but one were allowed an unrestricted diet (but provided with an emergency protocol of frequent feedings in case of acute illness). None of them have so far been symptomatic or shown significantly elevated CK levels, and they have not displayed echocardiographic signs of cardiomyopathy. The single symptomatic newborn with VLCADD presented with hypoglycemia and lactic acidosis at 20 hours of age and was supplemented with intravenous dextrose before and during the collection of the NBS sample (the C14:1 screening result was 2.0 µmol/L). This child was treated with a long-chain-fat-restricted diet and later showed normal development. Another five reported VLCADD cases were defined to be FP cases due to high VLCAD residual enzyme activity (rEA, 23%, 28%, 30%, 37% and 52%). The R4S/CLIR dual scatter plot classified these as “VLCADD cases”. One of the individuals had a C14:1 of 1.8 µmol/L in the initial screening sample, had 52% rEA and was *ACADVL* wild-type. The two cases with 23% and 30% rEA had single disease-causing alleles, NM_000018.3 (ACADVL):c.1837C>T and c.856G>A, respectively, combined with wild-type alleles. The remaining two cases (28% and 37% rEA) were compound heterozygotes including variants of uncertain significance: NM_000018.3(ACADVL):c.[481G>A];[1711G>A] and c.[495G>T];[848T>C], respectively. The only surviving TFP patient, who has remained asymptomatic, had ambiguous screening results and was classified as “possible LCHAD/TFP” by the R4S/CLIR tool. Genetic analysis showed two variants previously described as pathogenic (ID 78, Table 4). In this case, molecular testing identified a TP case that would otherwise not have been reported.

In four newborns, the screening results showed a typical acylcarnitine profile indicative of MADD/GA-II (Table 2). Two of these patients were symptomatic before the NBS results were available (Table 3). The first patient with transient riboflavin responsive MADD, previously published [16], was admitted, critically ill, at 4 days of age with severe transient hyperammonemia. The second symptomatic riboflavin responsive MADD/GA-II individual presented with cardiac arrythmia and hypoglycemia. The remaining two were FP cases.

#### 3.2.2. Aminoacidopathies

Two patients with classical MSUD were recalled from home after abnormal NBS results and had subtle symptoms upon admittance to hospital. Both had leucine levels >2500 µmol/L, received hemofiltration (Table 3) and were later recipients of a liver transplant at 2 and 6 years of age, respectively, both with good outcomes. Two patients with CBS deficiency were ascertained: a pyridoxine non-responsive case and a case with a partial pyridoxine response. The total homocysteine (tHcy) measured at follow-up prior to pyridoxine supplementation in the latter patient did not exceed 60 µmol/L and declined to 30 µmol/L upon pyridoxine treatment and an unrestricted diet. This patient had one mutation associated with pyridoxine responsiveness (ID 19, Table 4). Both patients showed normal development. Five cases of tyrosinemia type 1 had an uneventful outcome upon treatment with a protein-restricted diet and 2-(2-nitro-4-trifluoromethylbenzoyl)-1, 3-cyclohexanedione (NTBC) therapy.

#### 3.2.3. Organic Acidurias

Three patients with MMA (MUT^0^, MUT^−^, Cbl B), three cases with PA, one cobalamin C-deficient patient and two patients with IVA were true screening positives. Six of the nine patients were symptomatic by the time the NBS results were available (Table 3). In 14/56 (25%) of the cases reported for suspected MMA/PA, maternally derived B12 deficiency was revealed. Moreover, B12 deficiency was incidentally detected during follow-up testing in four FP cases reported for CTD (n = 2), BD and MADD. Isovalerylcarnitine (C5) was above the cut-off (> 1 µmol/L) in 5026 of the 461,369 screened samples (1:91). However, in only two cases did the second-tier test reveal C5 values consistent with a possible diagnosis of IVA. The remaining samples were pivaloylcarnitine (pivmecillinam) positive. No case of β–ketothiolase deficiency or HMG-CoA lyase deficiency was identified. A four-year-old child who presented with severe metabolic acidosis and coma was confirmed with symptomatic 3-MCC. Retrospectively, a high level of C5OH was detected in her screening analysis, but she was not reported as screening-positive since the Sanger sequencing of *HLCS* showed wild-type alleles. A single case of HCS (biotin responsive) was ascertained. Of the 13 infants with BD, five were severe (<10% residual activity), and the remaining cases had partial deficiency (>10% residual activity, Table 4). A likely pathogenic stop variant not previously reported, NM_000060.2(BTD):c.1006C>T, (p.Gln336*), was observed in a sibling pair (ID 35 and 36, Table 4) with severe BD. In partial BD cases, the variant c.1330G>C (p.Asp444His), known to be related to a mild phenotype, was detected in one of the alleles in most cases (Table 4). The threshold for initiating the treatment of partial BD based upon confirmatory testing was revised twice during 2012–2019: From 2012–2016, patients with biotinidase activity <30% received biotin treatment. This threshold was reduced twice; between 2016 and 2017 and from 2018 onwards, only patients with confirmed < 25% and < 20% biotinidase activity, respectively, were started on biotin treatment (15–20 mg/day for profound BD and 5–10 mg/day for the partial BD).

### 3.3. Missed Cases during eNBS 2012–2020

Five false-negative (FN) cases were recorded during this period: A toddler with intermittent MSUD was diagnosed during an episode of hypoglycemia and ketoacidosis at 18 months of age (plasma leucine, 1300 µmol/L). His 9-month-old asymptomatic sister had normal leucine levels when tested, but variant MSUD was confirmed by genetic analysis in both siblings (ID 17 and 18, Table 4). The original NBS analysis showed normal biochemical values well below the cut-off in both siblings: Xle, 44 µmol/L and 172 µmol/L (cut-off >250 µmol/L) and Xle/Ala, 0.62 and 0.58 (cut-off >1.3), respectively. Allo-isoleucine was also undetectable in the second-tier test in both cases (<2 nmol/L). The FN CPT-II case was a two-year-old boy admitted to hospital for an acute infection where increased transaminases and CK levels were detected at the initial workup. As part of further diagnostic evaluation, metabolic workup analysis showed a plasma acylcarnitine profile compatible with CPT-II/CACT, and the genotype confirmed a myopathic phenotype (ID 64, Table 4). A review of his screening data showed values below the NBS cut-off: a (C16 + C18:1)/C2 of 0.32 (cut-off >0.52) and C16 of 2.0 µmol/L (cut-off > 5.5 µmol/L). Two asymptomatic siblings with CTD were incidentally detected with low plasma free carnitine as part of a general metabolic workup performed due to an unrelated condition (macrocephaly) in one of the siblings. They were both homozygotic for a previously reported missense variant associated with a mild phenotype (ID 40 and 41, Table 4). Reexamining the original screening values revealed C0 4.6 µmol/L and C3 + C16 of 2.3 µmol/L in the index case and C0 6.7 µmol/L and C3 + C16 1.96 µmol/L in his sibling. Neither of these had been reported from NBS due to the combination of borderline and normal values in the primary screening markers. In these cases, the R4S/CLIR tool was not informative. The scores and percentiles (presented as medians and ranges) and total counts for all true and false positives, and false negatives, for every condition detected in Norway since 2012 are presented in Table 5. The table shows the disorders for which the CLIR/R4S tools are able to perform an important decision support role directly from the first-tier biochemical screen and, conversely, where second-tier analyses are still required in order to determine the true status of a sample.

These results do not represent the full discriminatory power of the CLIR tools—particularly the dual scatter plot—to further distinguish between VLCADD true positives and carriers, for example, but they clearly show the disorders for which the CLIR/R4S tools significantly support decision-making based on the first-tier screen. They also highlight where second-tier analyses are still required in order to determine the true status of a sample. In the case of CTD, for example, where high levels of antibiotic use causes significant numbers of false positives, we see that there is complete overlap between the percentile or score ranges for false positives and true positives. In this case, the standard R4S/CLIR algorithm, which is optimized for sites without antibiotic interference, is unable to help us to distinguish between the 23 false and three true positives. This interference means that many CTD-normal individuals receive a non-zero score and explains how false-negative results with non-zero scores were indistinguishable from normal results. By contrast, there is very little overlap between the range of scores for the MMA/PA true and false positives. Given the 56 false positives generated over the 8-year period, it is clear that the R4S/CLIR tools could reliably be used to classify the majority of these correctly from the first-tier screening result. MSUD is also a condition that exemplifies the R4S/CLIR tool’s capacity to correctly separate true and false positives by scores, although the two false-negative samples with absolutely normal leucine and valine values show that there will always be a few exceptional cases that no combination of screening tools based on biochemical markers is able to detect.

## 4. Discussion

We have described the implementation of the eNBS program for IEMs in Norway over eight years in terms of both methodological refinements and the interim evaluation of patient outcomes. The immediate introduction of 19 additional IEMs in 2012, without any previous experience in interpreting the borderline screening values and inherent methodological pitfalls associated with several of the screening conditions, resulted in a high number of FPs during the first year of eNBS. Nevertheless, despite the lack of preliminary pilot studies, the overall PPV (40%) was still comparable to that in other screening programs [18,19]. The combination of biochemical and molecular methods has resulted in a clear improvement in the PPV for these disorders from 26% in 2012 to 54% in 2019. The NBS Unit has evolved from being exclusively a biochemical screening laboratory to include diagnostic confirmation by rapid molecular analyses on DNA from dried blood spots with turnaround times of 1–3 working days. However, biochemical test modalities and enzyme analyses remain vital to identify, characterize, individualize and even stop treatment. After reporting a positive screening based on sequencing results, the utility of and requirement for parental segregation testing need to be emphasized, allowing for the clarification of cis and trans alleles, and the exclusion of allele drop-out and assumed deleterious variants located in cis in genes for autosomal recessive disorders.

The test to separate pivaloyl-carnitine from isovaleryl-carnitine was introduced within a month after the start of eNBS. Over an eight-year period, 5026 pivaloyl-carnitine (pivmecillinam)-positive cases flagged as IVA were not falsely reported as NBS disorders, and only two true IVA cases were reported and confirmed. The prescription of pivalic-containing antibiotics (pivmecillinam as a treatment for urinary tract infections) is prevalent among women in Norway [20]. As a secondary effect, the associated low free carnitine (C0) following pivmecillinam treatment confuses the interpretation of a low C0 (a primary marker of CTD), thereby hampering standard interpretation approaches (including CLIR), which depend on this marker. Another significant interference is the use of total parenteral nutrition (TPN), which affects the amino acid and acylcarnitine profiles of predominantly premature neonates. Notably, our first (symptomatic) IVA case (diagnosed before the NBS result, Table 3) was classified as negative using the R4S tool (2012) since the patient was receiving TPN at the time of the NBS sample collection.

Overall, three out of four FP cases were screened as positive for CTD and MMA/PA. As has been shown in a number of publications, screening for CTD remains challenging, with both low sensitivity and specificity [21,22]. Shortly after birth, carnitine levels reflect maternal levels, and CTD in the infant may easily be missed when using a cut-off-based algorithm unless a second test is performed [23]. We had two possible CTDs among many cases with borderline screening values that escaped reporting, and these were later diagnosed only by coincidence. On the other hand, besides the physiological low maternal levels encountered in the last trimester [24], free carnitine in the screening sample also reflects maternal CTD as demonstrated in our two maternal cases and may furthermore reveal other maternal IEMs with secondary low free carnitine levels. Several screening programs have reported a higher detection rate for (asymptomatic) maternal CTD cases than affected off-spring [18,19,22,25], and in New Zealand, NBS for CTD was removed from the screening program for this reason. In Norway, the introduction of second-tier molecular analysis has reduced both the number of CTD FPs and the detection of maternal CTD cases. Disease-causing variants can occur outside the coding regions, such as in deep intronic variants and promoter and intergenic variants, which are not studied by conventional DNA sequencing [26,27], so confirmatory molecular testing, commonly restricted to exonic regions, has its limitations. Technical and bioinformatic limitations can often make it difficult to detect copy number variants and other structural variants. Surprisingly, no patients with CTD have been detected clinically in Norway in the last 18 years. This is somewhat puzzling, as a portion of the Faroe Island population, known for its high incidence of CTD [28], descended from the western part of Norway and settled in the Faroe Island in 900AD [29]. The founder mutation NM_003060.3(SLC22A5):c.95A>G, responsible for the high incidence of CTD in the Faroe Island, was detected in 11 alleles (4%) of 132 screened samples sequenced for CTD. It is possible that symptomatic patients with CTD in Norway have escaped diagnosis, have been diagnosed abroad, or have died as early as the neonatal period without being diagnosed [30]. It is also plausible that cases detected by NBS with the missense variant NM_003060.3(SLC22A5):c.136C>T, (p.Pro46Ser) in one or both alleles have milder phenotypes or even could remain asymptomatic without treatment. Thus, screening for CTD remains challenging.

Propionylcarnitine (C3) is yet another challenging screening marker, and in accordance with other NBS publications, the overall PPV without second-tier testing was only about 10% for MMA and PA [18,19,31]. The effect of second-tier methods for MMA and methylcitrate (implemented in 2019) on PPV still needs to be established, as the number of samples subjected to second-tier testing for MMA/PA are currently very limited. Second-tier DNA testing for PA and MMA is currently not fast enough, as the reporting of these serious disorders is time sensitive and should not be delayed.

The median time needed to close an FP case was significantly greater than the time required to diagnose a TP case. This was mostly a concern in the first years of the eNBS program, with many families left in a state of uncertainty during the confirmatory process. Several factors may come into play. First, for the TP cases, the urgency of starting appropriate treatment and avoiding symptoms drove the prioritization of the diagnostic process, as well the subsequent close follow-up of the patient. In a number of the FP cases, local pediatricians were responsible for tracking the results of pending work-up, and this may have increased the waiting time for the families. Additionally, for some of the cases, enzyme analyses in lymphocytes and even fibroblasts were needed before a final conclusion could be drawn. In infants where diagnostic work-up revealed maternal B12 deficiency, a protocol with a year-long follow-up was recommended, and the last day of visit was noted as the day of closure of the false-positive case. Nevertheless, our impression of the fraction of FP cases admitted at our hospital was that parents expressed high levels of understanding and confidence towards the screening system even when their child was exposed to unnecessary testing, provided that follow-up was scheduled without delay and access to the metabolic team was given while waiting. In the last four years, the PPV has been 50% or higher for metabolic diseases, and the time needed to settle the FP cases has been substantially reduced, diminishing the burden of FP cases on both families and the health system.

Compared to children born in the previous decade (2002–2012), the overall incidence of conditions covered by NBS increased substantially, corresponding to the findings of others who have done similar comparisons [18,19,32,33]. The predominant explanation is the detection of increased numbers of milder phenotypes (in particular, BD, VLCADD and MCADD). However, eNBS has allowed earlier diagnosis and pre-symptomatic treatment for the severe phenotypes of the same conditions as well as for a number of other cases such as Tyr-I and GA-I as compared to in the pre-screening era, which gives a significant improvement in healthcare benefits. The incidence of MCADD nearly tripled after eNBS was started in Norway. Such an increase in MCADD incidence is in accordance with the experience of other screening programs [18,33,34]. The incidence of MCADD in Norway with eNBS (1:27,139) matches the report from the Czech Republic (MCADD incidence, 1:22,000) [35]. The incidence of MCADD after eNBS in Norway remains the lowest published in Western Europe, underscored by the fact that one third of our detected cases were born to immigrant parents. In Denmark and in the Netherlands, by comparison, the incidence of MCADD is the highest in Europe and almost threefold higher than that Norway (1:8954 and 1:8300, respectively) [34,36]. Initially, enzyme analysis in lymphocytes was undertaken in all MCADD patients including those homozygous for c.985A > G (n = 10), but as the latter genotype showed residual enzyme activity of 3% or lower for those tested, enzyme analysis was eventually omitted for the c.985A > G homozygous individuals. As demonstrated in a recent Dutch study, all the homozygous cases of c.985A > G tested showed low or no residual activity [34]. The only CPT-IA encountered was initially missed on confirmatory acylcarnitine analysis. CPT-IA (elevated free carnitine concentrations and low acylcarnitine concentrations) is more readily detected in the DBSs than plasma due to free carnitine accumulation in red blood cells [37]. The patient presented before DNA analysis in DBSs was established as a second-tier test, and the confirmation of the diagnosis required both a skin biopsy for fibroblast analysis and a blood sample to be sent abroad for DNA analysis; the completion of the diagnostic process took more than two months. Later, the mutations were confirmed by DNA analysis using the original DBS.

VLCADD is another important disorder in terms of the increased incidence of milder phenotypes that emerges through screening [38,39,40,41]. Enzyme analysis in lymphocytes for residual activity was used as a valuable tool to recognize carriers and biochemical phenotypes with variants of uncertain significance (VUS) alleles but also to guide treatment intensity [42,43]. That is, for the milder phenotypes with 10–20% residual enzyme activity, both diet and fasting hours were relaxed when healthy. That said, the follow-up of milder VLCADD patients required many referrals for prophylactic glucose polymer treatment during intercurrent illness, exacerbated in children who tended to be prone to infections during their first years of life. In total, FAO cases as a group (nine FAO conditions included in our eNBS) obtained a pooled incidence of only 1:11,800 due to the incidence of MCADD being lower than expected.

In line with several other screening programs, partial BD was more prevalent in our eNBS program than severe BD [35,44,45]. No patients were clinically ascertained as having BD in the decade before eNBS in Norway, and profound BD was only detected in non-European immigrants, which corresponds to the Swedish findings for BD screening [44], rendering profound BD an ultra-rare disease in the native population. Regardless, the benefit of BD screening is well established, considering that severe disease can be prevented by a simple vitamin supplement [45]. There is less consensus and evidence for the need to treat partial BD, as patients with break-through symptoms during infections or other stress have been reported only rarely [45,46]. Our experience has been that the families of children with partial BD demand the same follow-up health resources as those with severe phenotypes. Common symptoms such as other types of skin rashes that arise during infancy and early childhood were frequently linked to partial BD by the family or even by other health personnel, which created unnecessary anxiety in the families despite good compliance. In line with the practices of some other western European screening programs [44,47], we chose to adjust the cut-off for biotin treatment from 30% to 20% biotinidase activity.

Twenty-one percent of our TP cases presented symptoms at a median age of 2.5 days after birth (0–5 days), which is a higher proportion than that reported elsewhere [19,48,49,50], although not directly comparable, as screening programs slightly differ between countries. Vigilance by neonatal intensive care units should be emphasized, avoiding the waiting for screening results before action is taken in cases with a suspected metabolic disease. It is worth noting, however, that in the majority of these symptomatic cases (11/17), the NBS result was still the first mode of detection, which enabled rapid intervention and the initiation of correct treatment.

Refsum et al. predicted the birth prevalence of CBS deficiency in Norway to be 1:6400 births, based on the allele frequencies of six pathogenic bi-allelic mutations among 1133 NBS samples; more than 90% of the alleles were associated with a pyridoxine-responsive phenotype [51]. Given this estimate and full penetrance, about 70 newborns with a combination of these mutations should have been born during our 8 years of eNBS. Only two cases with CBS deficiency were detected (and only one of them had one of the six variants reported by Refsum et al.), and no false-negative cases have so far been brought to our attention throughout the time period of eNBS. However, re-analyzing the DBS of a child diagnosed with pyridoxine-responsive CBS born three years before eNBS showed a methionine concentration of only 27 µmol/L. It is well recognized, and our screening program is evidently no exception to the experience of others, that using methionine for first-tier screening may fail to detect pyridoxine-responsive CBS patients [52]. After tHcy testing was introduced as a second-tier test for CBS, followed by Sanger sequencing as third-tier, the reporting of other causes of hypermethioninemia not included in eNBS has been avoided. Whether lowering the cut-off for methionine and performing many more biochemical and molecular tests are cost-effective is not clear. The ethical aspect of detecting newborns with a very mild form of CBS who may remain asymptomatic or first develop symptoms in adulthood [53] must also be taken into account in the attempt to increase sensitivity.

The occurrence of false-negative cases is expected in any screening program [19,52]. It is well known that myopathic CPT-II [54,55] and variant MSUD [52,56] may be missed due to screening markers lying below the action limit. In Norway, the incidence of intermittent MSUD is more common than that of the classical phenotype [57], and it is likely that most intermittent MSUD cases will be missed using the existing screening algorithm. On the other hand, the two classical MSUD cases, which were detected with unrecognized subtle symptoms upon reporting, showed toxic levels of leucine upon admittance (Table 3), demonstrating the importance of NBS for early treatment.

We cannot exclude that some patients were not accounted for (i.e., died without diagnosis, were diagnosed but not registered, or even received a diagnosis abroad) during the decade prior to eNBS. A survey of physicians caring for metabolic patients in the other health regions of Norway did not reveal any patients who had escaped registration, so the observed number of patients is likely to be as complete as possible. Another limitation is that at present, no national metabolic patient registry has been established in Norway (with the exemption of that for NBS diseases) and precise data on patient outcomes from the decade prior to eNBS were not available. Therefore, a direct comparison to the cohort identified by eNBS could not be performed. Developmental and neuropsychiatric evaluation are also missing from the standardized part of our follow-up program for all patients detected by NBS, and developmental delay, particularly at the milder end of the spectrum, may pass unrecognized for the first few years of life. The cost/benefit ratio of introducing a routine neuropsychological assessment program remains debatable, particularly for the milder phenotypes detected exclusively by NBS, since there is a high probability of a normal outcome.

## 5. Conclusions

The overall performance of our eNBS for IEMs has improved significantly over the last eight years, accomplishing one TP case for every FP reported. In general, the biochemical profile including second-tier testing and supplemented by the interpretation provided by the R4S/CLIR in silico tools remains the strongest basis for the decision on whom to report. However, the rapid DNA analysis of DBSs has reduced the dependency on targeted biochemical diagnostics and, in cases with borderline screening values, prevented false-negative outcomes. On the other hand, the unresolved challenges of molecular testing are the interpretation of VUS and the process of determining when a negative DNA result should override a positive biochemical test. The NBS regulation in Norway mandates the supervision of the national NBS program and the evaluation of its outcomes. A multidisciplinary team with technical, biochemical, genetic, bioinformatic and clinical expertise located in the same hospital has enabled timely NBS adjustments. The opportunity for the participation of metabolic physicians in both the evaluation of screening results and the clinical follow-up of the majority of patients has given first-hand experience of the advantages and challenges of the national eNBS program for IEMs from both a technical/analytical and clinical point of view. The majority of cases picked up by NBS had favorable outcomes, benefitting from pre-symptomatic diagnosis, although the lives of a few newborns with a severe TFP phenotype could not be saved despite early treatment in accordance with the findings of Sperk et al. [58]. We also note as other before us [59] that the majority of patients with MMA/PA still have persistent morbidity despite screening (but 100% survival). As emphasized by others [32], an important spin-off effect of the expansion has been the national overview gained and increased competence of addressing metabolic disorders in general. This also includes secondary educational benefits and awareness-raising among frontline health professionals nationally, who are often the first point of contact for these patients. The further automatic and real-time evaluation of biochemical and genetic screening data with bioinformatics tools is the next step for improving our NBS program. The major challenge from a clinical and public health viewpoint is still the detection and follow-up of attenuated phenotypes (e.g., VLCADD, BD and MCADD) that pose a potential burden on families and the health care system. Joint action enabled by the R4S/CLIR program, the International Society of Newborn Screening (ISNS), the Society for the Study of Inborn Errors of Metabolism (SSIEM) and the European Reference Network for Hereditary Metabolic Disorders (MetabERN) is warranted for the design of NBS outcome studies and agreement on case definitions. This would support jurisdictions in their evaluations of which diseases/disease spectrum to include in their respective NBS programs.

## Figures and Tables

**Figure 1 IJNS-06-00051-f001:**
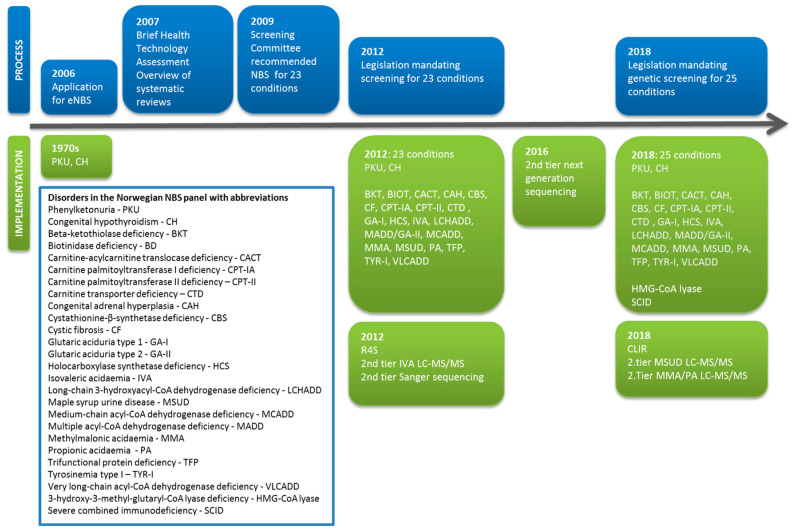
Timeline for the development of newborn screening (NBS) in Norway, showing the administrative and legislative process (blue) alongside the inclusion of new disorders and the implementation of new screening methods (green).

**Table 1 IJNS-06-00051-t001:** Screening thresholds, results values and start years for dried blood spot (DBS) molecular analyses in true and false positives reported during expanded newborn screening (eNBS) for 2012–2020.

Condition	Cut-Off Values	Screening Results Values (µmol/L or Ratio)		Molecular Analyses (Start Year)
	µmol/L or ratio	True-positive cases	False-positive cases	
		Median (range)	Median (range)	
MMA/PA	C3 > 4.75	14.1 (10.1–27)	8.9 (3.3–18.5)	2016 (NGS)
MMA/PA	C3/C2 > 0.25	0.74 (0.45–1.16)	0.27 (0.11–0.40)	
PA	C4/C3 < 0.05	0.02 (0.01–0.02)	0.05 (0.02–0.2)	
IVA	C5 > 1	8.2 (3.4–12.9)	-	2012
	C5/C0 > 0.04	0.9 (0.2–1.6)	-	
GA-I	C5DC > 0.4	4.25 (2.2–5.4)	-	2012
	C5DC/C16 > 0.1	1.12 (0.34–1.22)	-	
MSUD	LEU\ILE\PRO-OH > 250	1540 (1300–1790)	290 (284–296)	2016 (NGS)
	LEU\ILE\PRO-OH/ALA > 1.3	4.2 (0.84–7.6)	1.65 (1.55–1.75)	
CBS	MET > 40	93.8 (69.1–119.0)	78.3 (56.7–99.9)	2014
	MET/PHE > 0.7	1.74 (1.18–2.3)	1.23 (0.71–1.80)	
TYR-I	SUAC > 2	9.1 (7.5–16.2)	5.2 (3.1–5.4)	2013
HCS	C3 > 1.57	3.1 (3.1–3.1)	1.7(1.3–2.2)	2012
	C5OH > 0.85	2.38 (2.38–2.38)	5.49 (1.3–9.7)	
HMG-CoA lyase	C5OH > 0.85	-	-	2018
BKT	C5:1 > 0.1	-	-	2013
	C3DC\C4OH > 0.5	-	-	
BD	<60 U/dL	32 (7–58)	43 (31–57)	2013
CTD	C0 < 6	3.9 (1.7–6.2)	4.7 (2.5–22)	2013
	C3 + C16 > 2	1.48 (1.27–1.61)	1.94 (0.37–11.6)	
CPT-IA	C0/C16 + C18 > 40	277 (277–277)	41.8 (41.8–41.8)	2012
	C16 + C18:1/C2 < 0.15	0.14 (0.014–0.014)	0.098 (0.098–0.098)	
MCADD	C8 > 0.4	10.71 (2.4–28.0)	0.65 (0.41–0.69)	2012
CPT-II /CACT	C16 + C18:1/C2 > 0.52	1.42 (1.26–10.8)	0.44 (0.31–0.57)	2012
	C16 > 5.5	5.1 (4.64–13.1)	8.8 (7.8–9.8)	
VLCADD	C14:1 > 0.5	1.9 (1.0–2.0)	1.4 (1.0–2.0)	2012
	C14:1/C2 > 0.02	0.11 (0.03–0.18)	0.06 (0.04–0.07)	
LCHADD	C16OH > 0.1	1.6 (1.6–1.6)	0.14 (0.14–0.14)	2012
	C18OH > 0.1	0.92 (0.92–0.92)	0.35 (0.35–0.35)	
TFP	C16OH > 0.1	1.1 (.57–1.6)	-	
	C18OH > 0.1	0.81 (0.62–1.0)	-	
MADD/GA-II	C14:1/C2 > 0.02	0.07 (0.05–0.09)	0.1 (0.06–0.14)	2016 (NGS)
	C12 > 0.4	1.9 (1.9–1.9)	2.0 (2.0–2.1)	

DBS, dried blood spot; NGS, next generation sequencing; LEU, leucine; ILE, isoleucine; PRO-OH, hydroxyproline; ALA, alanine; MET, methionine; PHE, phenylalanine; SUAC, succinylacetone.

**Table 2 IJNS-06-00051-t002:** NBS results for 20 metabolic conditions in an eight year period from 2012 to 2020 compared to clinically presenting cases born between 2002 and 2012.

	2002–2012		eNBS 2012–2020		
	Clinically Presenting Cases		True-Positive Cases		False-Positive Cases
	n	Incidence	n	Incidence	n
MMA	5	1:119,318	4 *	1:115,342	56
PA	-	-	3	1:153,789	
IVA	4	1:149,147	2	1:230,684	-
GA-I	6	1:99,431	4	1:115,342	-
MSUD	5	1:119,318	2	1:230,684	2
CBS	1	1:596,591	2	1:230,684	2
TYR-I	8	1:74,573	5	1:92,273	3
HCS	2	1:298,295	1	1:461,369	2
HMG-CoA lyase	-	-	-^a^	-	-
BKT	1	1:596,591	-	-	-
BD	-	-	13	1:35,489	5
CTD	-	-	3	1:153,789	22
CPT-IA	-	-	1	1:461,369	1
MCADD	8	1:74,573	17	1:27,139	4
CPT-II	-	-	4	1:115,342	2
CACT	2	1:298,295	1	1:461,369	-
VLCADD	1	1:596,591	6	1:76,894	5
LCHADD	4	1:149,147	1	1:461,369	1
TFP	3	1:198,863	4	1:115,342	-
MADD/GA-II	2	1:298,295	2	1:230,684	2
Total number	52	1:11 472	75	1:6151	107

* Includes one case with cobalamin C deficiency. ^a^ Included in eNBS in 2018. MMA; Methylmalonic aciduria, PA; Propionic aciduria, IVA; Isovaleric aciduria, GA-I; Glutaric aciduria type I, MSUD; Maple syrup urine disease, CBS; Cystathionine-β-Synthase deficiency, Tyr-I; Tyrosinemia type I, HCS; Holocarboxylase synthetase deficiency, HMG-CoA-lyase; 3-hydroxy 3-methylglutaryl-CoA lyase deficiency, BKT; Beta-ketothiolase-deficiency, BD; Biotinidase deficiency, CTD; Carnitine transporter deficiency, CPT-IA; Carnitine palmitoyl-transferase-IA deficiency, MCADD; Medium chain acyl-CoA dehydrogenase deficiency, CPT-II; Carnitine palmitoyl-transferase-II deficiency, CACT; Carnitine-acylcarnitine translocase deficiency, VLCADD; Very long-chain acyl-CoA dehydrogenase deficiency; LCHADD; Long-chain 3-hydroxyacyl–CoA dehydrogenase deficiency, TFP; Trifunctional protein deficiency; MADD/GA-II; Multiple acyl-CoA dehydrogenase deficiency/Glutaric aciduria type II.

**Table 3 IJNS-06-00051-t003:** Newborns presenting with symptoms (*n* = 16/75) before NBS results were available.

Condition	Age at Presentation (Days)	Age at Final Diagnosis (Days)	Clinical and Biochemical Findings	Mode of First Detection
MMA	2	2	Lethargy, metabolic acidosis, hyperammonemia (260 µmol/L)	TD
MMA	2	6	Encephalopathy, hypoglycemia, metabolic acidosis, bulging fontanel hyperammonemia (1400 µmol/L)	NBS
PA	3	4	Encephalopathy, vomiting, metabolic acidosis, hyperammonemia (372 µmol/L)	TD
PA	3	4	Encephalopathy, vomiting, metabolic acidosis, hyperammonemia (740 µmol/L)	TD
PA	3	4	Encephalopathy, metabolic acidosis, seizures, hyperammonemia (1400 µmol/L)	NBS
IVA	3	4	Encephalopathy, metabolic acidosis, seizures, hyperammonemia (769 µmol/L)	TD
MSUD	3	5	Encephalopathy, abnormal movements, seizures (leucine 2560 µmol/L)	NBS
MSUD	5	5	Subtle encephalopathy, abnormal movements (leucine 2200 µmol/L)	NBS
CPT-IA	1	74 *	Hypoglycemia, lactic acidosis	NBS
MCADD	2	6	Severe hypoglycemia (p-glucose 0.1 mmol/l) with MRI correlate	NBS
CPT-II	0	7	Multi-organ failure from birth (microgyria, renal failure, cardiomyopathy)	NBS
VLCADD	1	6	Hypoglycemia, lactic acidosis, CK 10 000 U/L at 24 h of age.	NBS
TFP	1	9	Heart failure (dilated cardiomyopathy), respiratory distress	NBS
TFP	0	5	Heart failure (dilated cardiomyopathy), respiratory distress	TD
MADD/GA-II	4	704	Encephalopathy, metabolic acidosis, respiratory distress, hyperammonemia (740 µmol/L)	NBS
MADD/GA-II	1	4	Lethargy, hypoglycemia, lactic acidosis, hypoglycemia, arrhythmia	NBS

TD; targeted diagnostics, NBS; newborn screening, * Age at diagnosis for CPT-IA depended on fibroblast culture and enzyme analysis.

**Table 4 IJNS-06-00051-t004:** Results of genetic and enzymatic analyses in 75 true-positive cases and 5 false-negative cases.

ID	Enzyme	Condition	Reference Sequence	Allele 1	ACMG	Allele 2	ACMG
Organic acidurias							
1	F Cbl ^a^	MMA	NM_052845.3(MMAB)	c.291–1G > A (splice defect)	5	c.571C > T (p.Arg191Trp)	5
2	F MUT ^b^	MMA	NM_000255.4(MMUT)	**c.675_677delTAT (p.Phe225_Met226delinsLeu)**	5	c.1106G > A (p.Arg369His)	5
3	NP	MMA	NM_000255.4(MMUT)	c.1655C > T (p.Ala552Val)	5	c.1677-1G > A (splice defect)	5
4	NP	Cbl C	NM_015506.2(MMACHC)	c.271dupA (p.Arg91Lysfs*14)	5	c.271dupA (p.Arg91Lysfs*14)	5
5	F ^c^	PA	NM_000532.4(PCCB)	c.319G > A (p.Val107Met)	4	**c.1281_1282delCA (p.Thr428Serfs*12)**	5
6	NP	PA	NM_000532.4(PCCB)	c.331C > T (p.Arg111 *)	5	c.838dup (p.Leu280Profs*11)	5
7	NP	PA	NM_000532.4(PCCB)	c.1498 + 2T>C (splice defect)	5	c.1498 + 2T>C (splice defect)	5
8	L ^d^	IVA	NM_002225.3(IVD)	**c.208G>T (p.Glu70 *)**	5	c.941C>T (p.Ala314Val)	5
9	NP	IVA	NM_002225.3(IVD)	**c.296-2A>G (splice defect)**	4	**c.296-2A > G (splice defect)**	4
10	NP	GA-I	NM_000159.2(GCDH)	c.572T>C (p.Met191Thr)	5	c.1045G>A (p.Ala349Thr)	5
11	NP	GA-I	NM_000159.3(GCDH)	c.1045G>A (p.Ala349Thr)	5	c.1204C>T (p.Arg402Trp)	5
12	NP	GA-I	NM_000159.3(GCDH)	c.1240G>A (p.Glu414Lys)	5	c.1240G>A (p.Glu414Lys)	5
13	NP	GA-I	NM_000159.3(GCDH)	c.1240G>A (p.Glu414Lys)	5	c.1240G>A (p.Glu414Lys)	5
14	NP	HCS	NM_000411.6(HLCS)	c.1519 + 5G > A(splice defect)	5	c.1993C > T (p.Arg665 *)	5
**Aminoacidopathies**							
15	4.5% ^e^	MSUD		ND		ND	
16	NP	MSUD	NM_000709.3(BCKDHA)	c.375 + 648_484 + 520del p.Gly126Valfs*3 (ref below)	5	c.375 + 648_484 + 520del p.Gly126Valfs*3 (ref below)	5
17	NP	MSUD ^Ɨ^	NM_001918.3(DBT)	c.901C>T (p.Arg301Cys)	5	c.1291C>T (p.Arg 431 *)	5
18	NP	MSUD ^Ɨ^	NM_001918.3(DBT)	c.901C>T (p.Arg301Cys)	5	c.1291C>T (p.Arg 431 *)	5
19	P ^f^	CBS	NM_000071.2(CBS)	**c.451 + 2T>G (splice defect)**	4	c.833T>C (p.Ile278Thr)	5
20	NP	CBS	NM_000071.2(CBS)	**c.728A>G (p.Gln243Arg)**	4	**c.728A>G (p.Gln243Arg)**	4
21	NP	TYR-I	NM_000137.2(FAH)	c.554-1G>T (splice defect)	5	c.1062 + 5G>A (splice defect)	5
22	NP	TYR-I	NM_000137.2(FAH)	c.742delG (p.Pro249Hisfs*55)	5	c.1062 + 5G>A (splice defect)	5
23	NP	TYR-I	NM_000137.2(FAH)	c.742delG (p.Pro249Hisfs*55)	5	c.1062 + 5G>A (splice defect)	5
24	NP	TYR-I	NM_000137.2(FAH)	**c.1008C>G (p.Asn336Lys)**	4	c.1062 + 5G>A (splice defect)	5
25	NP	TYR-I	NM_000137.2(FAH)	c.1062 + 5G>A (splice defect)	5	c.1062 + 5G>A (splice defect)	5
26	S16% ^g^	BD	NM_000060.2(BTD)	c.235C>T (p.Arg79Cys)	5	c.1330G>C (p.Asp444His)	5
27	S22% ^g^	BD	NM_000060.2(BTD)	c.278A>G (p.Tyr93Cys)	5	c.1330G>C (p.Asp444His)	5
8	S2% ^g^	BD	NM_000060.2(BTD)	c.424C>A (p.Pro142Thr)	4	c.424C>A (p.Pro142Thr)	4
29	S11% ^g^	BD	NM_000060.3(BTD)	c.470G>A (p.Arg157His)	5	c.470G>A (p.Arg157His)	5
30	S18% ^g^	BD	NM_000060.3(BTD)	c.470G>A (p.Arg157His)	5	c.1330G>C (p.Asp444His)	5
31	S6% ^g^	BD	NM_000060.2(BTD)	c.470G>A (p.Arg157His)	5	c.1333G>A (p.Gly445Arg)	5
32	S23% ^g^	BD	NM_000060.3(BTD)	c.511G>A (p.Ala171Thr)	5	c.1330G>C (p.Asp444His)	5
33	S15% ^g^	BD	NM_000060.3(BTD)	c.511G>A (p.Ala171Thr)	5	c.1330G>C (p.Asp444His)	5
34	S11% ^g^	BD	NM_000060.3(BTD)	c.605A>T (p.Asn202Ile)	5	c.605A>T (p.Asn202Ile)	5
35	S7% ^g^	BD	NM_000060.2(BTD)	**c.1006C > T (p.Gln336 *)**	5	**c.1006C > T (p.Gln336 *)**	5
36	S < 1% ^g^	BD	NM_000060.2(BTD)	**c.1006C > T (p.Gln336 *)**	5	**c.1006C > T (p.Gln336 *)**	5
37	S27% ^g^	BD	NM_000060.3(BTD)	c.1330G>C (p.Asp444His)	5	c.1368A>C (p.Gln456His)	5
38	S8% ^g^	BD	NM_000060.2(BTD)	c.626G>A (p.Arg209His)	5	c.1595C>T (p.Thr532Met)	5
**Fatty acid oxidation defects**							
39	NP	CTD	NM_003060.3(SLC22A5)	c.51C>G (p.Phe17Leu)	5	c.136C>T (P.Pro46Ser)	5
40	F12% ^h^	CTD ^Ɨ^	NM_003060.3(SLC22A5)	c.136C>T (p.Pro46Ser)	5	c.136C>T (p.Pro46Ser)	5
41	NP	CTD ^Ɨ^	NM_003060.3(SLC22A5)	c.136C>T (p.Pro46Ser)	5	c.136C>T (p.Pro46Ser)	5
42	NP	CTD	NM_003060.3(SLC22A5)	c.136C>T (p.Pro46Ser)	5	c.844C>T (p.Arg282 *)	5
43	NP	CTD	NM_003060.3(SLC22A5)	c.847T>A (p.Trp283Arg)	4	c.847T>A (p.Trp283Arg)	4
44	F ^i^	CPT-IA	NM_001031847.2(CPT1A)	**c.619C > T (p.Gln207 *)**	5	**c.2215A > G (p.Lys739Glu)**	4
45	14% ^j^	MCADD	NM_000016.4(ACADM)	c.250C>T (p.Leu84Phe)	5	c.985A>G (p.Lys329Glu)	5
46	11% ^j^	MCADD	NM_000016.4(ACADM)	c.250C>T (p.Leu84Phe)	5	c.985A >G (p.Lys329Glu)	5
47	<1% ^j^	MCADD	NM_000016.4(ACADM)	c.362C>T (p.Thr121Ile)	5	c.362C>T (p.Thr121Ile)	5
48	6% ^j^	MCADD	NM_000016.4(ACADM)	c.388-19T > A (intronic)	5	c.985A>G (p.Lys329Glu)	5
49	NP	MCADD	NM_000016.4(ACADM)	c.985A>G (p.Lys329Glu)	5	c.985A>G (p.Lys329Glu)	5
50	NP	MCADD	NM_000016.4(ACADM)	c.985A>G (p.Lys329Glu)	5	c.985A>G (p.Lys329Glu)	5
51	NP	MCADD	NM_000016.4(ACADM)	c.985A>G (p.Lys329Glu)	5	c.985A>G (p.Lys329Glu)	5
52	NP	MCADD	NM_000016.4(ACADM)	c.985A>G (p.Lys329Glu)	5	c.985A>G (p.Lys329Glu)	5
53	<1% ^j^	MCADD	NM_000016.4(ACADM)	c.985A>G (p.Lys329Glu)	5	c.985A>G (p.Lys329Glu)	5
54	<1% ^j^	MCADD	NM_000016.4(ACADM)	c.985A>G (p.Lys329Glu)	5	c.985A>G (p.Lys329Glu)	5
55	3% ^j^	MCADD	NM_000016.4(ACADM)	c.985A>G (p.Lys329Glu)	5	c.985A>G (p.Lys329Glu)	5
56	<1% ^j^	MCADD	NM_000016.4(ACADM)	c.985A>G (p.Lys329Glu)	5	c.985A>G (p.Lys329Glu)	5
57	<1% ^j^	MCADD	NM_000016.4(ACADM)	c.985A>G (p.Lys329Glu)	5	c.985A>G (p.Lys329Glu)	5
58	NP	MCADD	NM_000016.4(ACADM)	c.985A>G (p.Lys329Glu)	5	c.985A>G (p.Lys329Glu)	5
59	<1% ^j^	MCADD	NM_000016.4(ACADM)	c.985A>G (p.Lys329Glu)	5	**c.1171A>G (p.Met391Val)**	3+
60	NP	MCADD	NM_000016.4(ACADM)	c.244dup (p.Trp82Leufs*23)	5	c.244dup (p.Trp82Leufs*23)	5
61	NP	MCADD	NM_000016.4(ACADM)	c.244dup (p.Trp82Leufs*23)	5	c.244dup (p.Trp82Leufs*23)	5
62	L ^k^	CPT-II	NM_000098.2(CPT2)	c.149C>A (p.Pro50His)	5	c.149C>A (p.Pro50His)	5
63	L ^k^	CPT-II	NM_000098.2(CPT2)	c.149C>A (p.Pro50His)	5	c.1369A>T (p.Lys457 *)	5
64	NP	CPT-II ^Ɨ^	NM_000098.2(CPT2)	c.338C>T (p.Ser113Leu)	5	c.481C>T (p.Arg161Trp)	4
65	L ^k^	CPT-II	NM_000098.2(CPT2)	c.338C>T (p.Ser113Leu)	5	**c.1444_1447del (p.Thr482Trpfs*49)**	5
66	NP	CPT-II	NM_000098.2(CPT2)	**c.1798G>A (p.Gly600Arg)**	4	**c.1798G>A (p.Gly600Arg)**	4
67	F ^l^	CACT	NM_000387.5(SLC25A20)	**c.82G>T (p.Gly28Cys)**	5	**c.82G>T (p.Gly28Cys)**	5
68	15% ^m^	VLCADD	NM_000018.3(ACADVL)	c.533T>C (p.Leu178Pro)	5	c.1066A>G (p.Ile356Val)	3
69	17% ^m^	VLCADD	NM_000018.3(ACADVL)	c.848T>C (p.Val283Ala)	5	c.848T>C (p.Val283Ala)	5
70	12% ^m^	VLCADD	NM_000018.3(ACADVL)	c.848T>C (p.Val283Ala)	5	c.848T>C (p.Val283Ala)	5
71	7% ^m^	VLCADD	NM_000018.3(ACADVL)	c.848T>C (p.Val283Ala)	5	c.865G>A (p.Gly289Arg)	5
72	9% ^m^	VLCADD	NM_000018.3(ACADVL)	c.848T>C (p.Val283Ala)	5	**c.1177A>G (p.Thr393Ala)**	3+
73	<1% ^m^	VLCADD	NM_000018.3(ACADVL)	c.1837C>T (p.Arg613Trp)	5	c.1837C>T (p.Arg613Trp)	5
74	NP	LCHADD	NM_000182.4(HADHA)	c.1528G>C (p.Glu510Gln)	5	c.1528G>C (p.Glu510Gln)	5
75	L ^n^	TFP	NM_000182.4(HADHA)	c.1678C>T (p.Arg560 *)	5	c.1678C>T (p.Arg560 *)	5
76	NP	TFP	NM_000182.4(HADHA)	c.1678C>T (p.Arg560 *)	**5**	c.1678C>T (p.Arg560 *)	5
77	NP	TFP	NM_000183.2(HADHB)	**c.209 + 1G>C (splice defect)**	**5**	**c.255-1G>A (splice defect)**	5
78	L ^o^	TFP	NM_000182.4(HADHA)	c.180 + 3A>G (splice defect)	**5**	c.180 + 3A>G (splice defect)	5
79	F ^p^	MADD/GA-II	NM_017986.3(SLC52A1)	**c.1134 + 11G>A (intronic)**	**1**	**wild type**	
80	F22% ^q^	MADD/GA-II	NM_000126.3(ETFA)	**c.348A>T (splice defect)**	**3+**	**c.348A>T (splice defect)**	3+

New pathogenic DNA variants are shown in **bold**, and variants are classified according to the ACMG criteria: 5 for pathogenic, 4 for likely pathogenic, and 3 for variant of unknown significance [10]. ^Ɨ^ False-negative cases; FCblB ^a^, deficient14C propionate incorporation in fibroblasts (Cbl B complementation group); FMUT ^b^, deficient 14C propionate incorporation in fibroblasts (MUT0 complementation); NP, enzyme analysis not performed; F ^c^, deficient propionyl-CoA carboxylase activity in fibroblasts at 0.02 nmol/min.mg, reference range 0.42–2.6; L ^d^, isovaleryl-CoA dehydrogenase activity in lymphocytes <0.14 nmol/min.mg, reference range 0.89–2.13; F ^e^, decreased decarboxylation rate in fibroblasts of [1–14C] leucine (4.5% of intra-assay control); P ^f^, plasma cystathionine-β-synthase (CBS) activity at 29.3 nmol/L (reference 100–1000), classified in vitro as a possible pyridoxine non-responder with some residual CBS activity; S% ^g^, serum biotinidase enzyme activity as percentage of the reference mean; F ^h^, plasma membrane carnitine transporter (OCTN2) activity in fibroblasts (12% of the reference mean); F ^i^, carnitine palmitoyltransferase I (CPT-I) activity in fibroblasts below limit of quantification of assay (<0.04 nmol/(min.mg protein), reference range 0.75–2.23; L ^j^, medium-chain acyl-CoA-dehydrogenase (MCAD) activity in lymphocytes <1–14% of the mean of reference values; L ^k^, carnitine palmitoyltransferase II (CPT-II) activity in leukocytes <0.5 nmol/mg prot./min, reference range 7–12 nmol/mgprot./min; F ^l^, mitochondrial carnitine/acylcarnitine transporter activity below the limit of the quantification of the assay (<8 pmol/(min.mg protein), reference range 70–274; L ^m^, very long-chain acyl-CoA dehydrogenase activity in lymphocytes <1–17% of the mean of reference values; L ^n^, long-Chain 3- hydroxy-acyl-CoA dehydrogenase (LCHAD) activity in lymphocytes 15% of the reference mean and long-chain 3-ketoacyl-CoA thiolase activity 6% of the reference mean; L ^o^, long-chain 3-hydroxy-acyl-CoA dehydrogenase (LCHAD) activity in lymphocytes at 33% of the reference mean and long-chain 3-ketoacyl-CoA thiolase activity at 27% of the reference mean; F ^p^, normal overall beta oxidation (acylcarnitine profiling) in fibroblasts; F22% ^q^, oleate beta-oxidation activity in fibroblasts (flux assay) at 22% of the activity in controls. Ref ID 16, BCKDHA variant [17].

**Table 5 IJNS-06-00051-t005:** Region 4 Stork post-analytical interpretive tool (R4S)/Collaborative Laboratory Integrated Reports 2.0 (CLIR) results for 75 true-positive, 107 false-positive and five false-negative cases during eNBS for 2012–2020.

Condition	False Negatives		False Positives		True Positives	
BIOT			5 samples	percentiles: 22 (15–25)	13 samples	percentiles: 27 (16–86)
				scores: 128 (47–170)		scores: 188 (54–472)
CPT-IA			1 sample	percentiles: 0 (0–0)	1 sample	percentiles: 57 (57–57)
				scores: 0 (0–0)		scores: 553 (553–553)
CPT-II/CACT	1 sample	percentiles: 0 (0–0)	2 samples	percentiles: 0 (0–0)	5 samples	percentiles: 9 (0–82)
		scores: 1 (1–1)		scores: 11.5 (0–23)		scores: 166 (1–754)
CUD	2 samples	percentiles: 27 (19–35)	23 samples	percentiles: 39 (0–96)	3 samples	percentiles: 53 (38–88)
		scores: 70.5 (54–87)		scores: 97 (0–259)		scores: 128 (95–213)
GA-I					4 samples	percentiles: 61 (27–67)
						scores: 461 (175–504)
GA-II			2 samples	percentiles: 37.5 (36–39)	2 samples	percentiles: 40 (38–42)
				scores: 404 (370–439)		scores: 452 (415–490)
CBS			2 samples	percentiles: 13.5 (0–27)	2 samples	percentiles: 18 (4–32)
				scores: 71.5 (18–125)		scores: 108 (32–184)
IVA					2 samples	percentiles: 61 (46–76)
						scores: 416 (286–546)
LCHADD/TFP			1 sample	percentiles: 0 (0–0)	5 samples	percentiles: 34 (2–87)
				scores: 39 (39–39)		scores: 459 (132–760)
MCADD			4 samples	percentiles: 1.5 (0–2)	16 samples	percentiles: 81.5 (25–99)
				scores: 14.5 (3–24)		scores: 888 (265–1010)
HCS			2 samples	percentiles: 50 (0–100)	1 sample	percentiles: 27 (27–27)
				scores: 331 (83–579)		scores: 180 (180–180)
MMA/PA/CblC			56 samples	percentiles: 13 (0–29)	7 samples	percentiles: 31 (24–91)
				scores: 91.5 (0–252)		scores: 281 (194–683)
MSUD	2 samples	percentiles: 0 (0–0)	2 samples	percentiles: 10 (6–14)	2 samples	percentiles: 57 (37–77)
		scores: 0 (0–0)		scores: 60.5 (36–85)		scores: 393 (255–531)
TYR-I			3 samples	percentiles: 20 (4–20)	5 samples	percentiles: 23 (20–65)
				scores: 83 (14–83)		scores: 102 (79–195)
VLCADD			5 samples	percentiles: 20 (5–34)	6 samples	percentiles: 52 (11–66)
				scores: 129 (63–214)		scores: 312 (94–418)

In this table, we present post hoc CLIR/R4S single condition tool scores and the associated percentiles for all true-positive, false-positive and false-negative calls from the Norwegian screening program between 2012 and 2020. The data in each panel show the number of samples found, and the medians and ranges for the CLIR percentiles and CLIR scores for these individuals. Since the ranges of the CLIR scores are specific to each disorder, they cannot be used as a proxy for severity when comparing disorders. For this reason, we have also provided the CLIR percentiles, which provide normalization, allowing a greater degree of comparability.

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
