# Peer review of "Performance of Expanded Newborn Screening in Norway Supported by Post-Analytical Bioinformatics Tools and Rapid Second-Tier DNA Analyses"

_2409-515X, 2020, doi:10.3390/ijns6030051_

Round 1

Reviewer 1 Report

Thank you for this very interesting work and the opportunity to review it! While I recognize you have a tremendous amount of data en results to report, I do think the piece needs to be significantly shortened to ensure the readability; for example moving information to a supplementary file, a thorough check on double information in the text and tables, and double information in the discussion as well. The manuscript would also benefit from more structure; subheadings will help to make sure that all information pertaining a certain topic is put in the correct place and easy to follow for the readers. I do applaud your work and your efforts to put this into a publication.

I have added some suggestions in the PDF-file.

Reviewer 2 Report

The authors should be commended for the profound amount and quality of the work they have done. This paper is a fine example of a comprehensive presentation of the expansion of NBS - adapting and using current technologies and to improve their system proactively. Their ability to do this in their country is something that will be very hard to replicate. There is a lot of data presented here, and it can be a bit overwhelming for the reader, but this data is presented in methodical and logical manner of a very thorough description of NBS collection, second tier testing, confirmatory testing and outlier cases where there were exceptional circumstances. The separation of PKU was an excellent idea so to remove bias from the volume of these cases.

Data is comprehensive, and perhaps a bit overwhelming to read. Would it be possible to graphically re-present some of the data in a flow-diagram or chart?

This paper does suffer from many, many minor typographical errors – extra spaces or missing spaces between numbers and units, or spaces before or after hyphens, or between symbols and numbers (ie, "> 2500" should be ">2500", and "<8pmol" should be "<8 pmol", or “Cystathionine -β-Synthase” should be “Cystathionine-β-Synthase”). A few are listed below as well. 

Many abbreviations are used, and for the respective disorders – perhaps the authors could build a “master abbreviation” list at the beginning of the manuscript, then they can not repeat them in the multiple places in the text and tables and legends.

“cis” and “trans” should be in italics.

Line 292: you state TP confirmation was 2-704 days (and line 296) -but the MADD case took 705 days (line 295 and table 3). Please clarify or correct.

Line 256: typo “US’ ” – perhaps spell out United States of America

Table 2: why does the 4 have and * after in in the first row with MMA? And the superscript “a” with HMG-CoA lyase?

Table 3: Mis-spelling “hyperammonia” on Table 3 row 2 for MMA

Line 474: (Table 3) should be capitalized

Line 487: there is a missing start “(” parenthesis.

Table 5: The formatting of this table within the columns is very difficult to read. Please provide explanation for how the data is presented. E.g., for MCADD has 16 samples TP. So, is the “percentiles” 81.5 is an average or mean or median? And (25-99) is a min-max or 25%-75% range? Similarly, for “scores” – what are 888 and (265-1010).  This analysis and R4S/CLIR analysis could likely be an entire independent manuscript.

Lines 595-596: This sentence is a bit confusing, can you rephrase it?  “Large and small copy number variants, or other structural variants are not easily detected either.”

Line 663: you need a space between “with10-20%”

Author Response

Please see the attchment
